# Reporting Key Features in Cold-Adapted Bacteria

**DOI:** 10.3390/life8010008

**Published:** 2018-03-13

**Authors:** Paula M. Tribelli, Nancy I. López

**Affiliations:** 1Departamento de Química Biológica, Facultad de Ciencias Exactas y Naturales, Universidad de Buenos Aires, C1428EGA Buenos Aires, Argentina; paulatrib@qb.fcen.uba.ar; 2IQUIBICEN, CONICET, C1428EGA Buenos Aires, Argentina

**Keywords:** psychrophile, energy generation, polyhydroxyalkanoates, cell envelopes, cold-adapted bacterial communities

## Abstract

It is well known that cold environments are predominant over the Earth and there are a great number of reports analyzing bacterial adaptations to cold. Most of these works are focused on characteristics traditionally involved in cold adaptation, such as the structural adjustment of enzymes, maintenance of membrane fluidity, expression of cold shock proteins and presence of compatible solutes. Recent works based mainly on novel “omic” technologies have presented evidence of the presence of other important features to thrive in cold. In this work, we analyze cold-adapted bacteria, looking for strategies involving novel features, and/or activation of non-classical metabolisms for a cold lifestyle. Metabolic traits related to energy generation, compounds and mechanisms involved in stress resistance and cold adaptation, as well as characteristics of the cell envelope, are analyzed in heterotrophic cold-adapted bacteria. In addition, metagenomic, metatranscriptomic and metaproteomic data are used to detect key functions in bacterial communities inhabiting cold environments.

## 1. Introduction

Temperature is a key factor for bacterial survival. Cold environments are predominant over the Earth. A great amount of information was generated analyzing bacterial adaptations to cold (e.g., [1,2,3,4,5,6]). Traditional characteristics involved in cold adaptation entail structural adjustment of enzymes, maintenance of membrane fluidity, expression of cold shock proteins, adaptation of the translation and transcription machinery and the presence of compatible solutes [3].

Microorganisms inhabiting cold environments are denominated psychrophiles, a term defined by Morita [7] and proposed for microorganisms that have optimum growth temperatures lower than 15 °C with upper temperatures around 20 °C. From an ecological point of view, the names ‘eurypsychrophile’ and ‘stenopsychrophile’ were used to reflect wide and narrow tolerance to temperature, respectively [8]. Since growth rate is not the best indicator of physiological state under low temperatures [6,9], the degree of adaptation to cold should be considered for psychrophile definition. Different nomenclature can be used to describe different aspects related to life in cold conditions, however the term cold-adapted bacteria will be used in this work focused on bacteria indigenous to cold environments.

The aim of this review was to analyze different strategies of cold-adapted bacteria, looking for novel features and/or activation of non-classical metabolisms for a cold lifestyle. We propose that, superimposed on generalized characteristics—almost universally present in cold-adapted bacteria—there are other specific features of each taxonomic group that are relevant and contribute to the adjustment of the whole cold adaptation landscape. For this purpose, the contribution of metagenomic and metaproteomic data was analyzed to detect key functions in bacterial communities that inhabit cold environments.

## 2. Metabolic Features Related to Energy Generation in Cold Environments

Novel ‘multi-omic’ approaches allow the acquisition of information related to the functionality of genotypic traits. Most of these works involving transcriptomic and proteomic analysis have highlighted the modifications of metabolic pathways that take place under cold conditions. As a general conclusion from this amount of information, it is known that many studied bacteria downregulate primary metabolism under cold conditions (Table 1).

Oxygen solubility increases at low temperatures, generating the increase of reactive oxygen species (ROS), which leads to oxidative stress. For this reason, oxidative metabolism such as glycolysis, the pentose phosphate pathway, the tricarboxylic acid cycle (TCA) and the electron transport chain are generally depressed at low temperatures [19]. Although this adjustment of ROS-producing pathways is recognized, the underlying molecular mechanisms and the alternative metabolic pathways involved in bacterial cold-adaptation are still not fully understood. In this section, several selected examples of heterotrophic bacteria isolated from different cold environments are analyzed, trying to find support for the statement that cold-adapted bacteria can use shortened or non-central metabolic pathways to thrive in the cold.

Reconstruction of carbon metabolism in *Psychrobacter arcticus* 273-4, a Siberian permafrost bacterium able to grow at −10 °C, showed that it lacks glycolysis genes and a phosphotransferase system but possesses gluconeogenic enzymes, fructose-1,6-bisphosphatase and phosphoenolpyruvate synthase, suggesting that although it is not able to utilize sugars, other oxidized carbon sources are preferred [17]. Phenotypic analysis indicated that the energy metabolism of this bacterium is based on acetate, a compound found in that environment that can easily diffuse into the cell without cost-associated transport systems. A recent study of transcriptomic and proteomic responses to low temperatures in *Psychrobacter* sp. PAMC 21119, isolated from Antarctic permafrost soil, also showed that pathways for acetyl-CoA metabolism were upregulated while proteins involved in energy production and conversion were downregulated [20]. The utilization of glyoxylate shunt can provide intermediate carbon compounds to fulfill biosynthetic requirements [17]. The induction of genes encoding enzymes involved in the glyoxylate cycle at 5 °C was also observed in *Nesterenkonia* sp. AN1, a member of *Actinobacteria* isolated from Antarctic soil [11]. In another bacterium isolated from permafrost—*Exiguobacterium sibiricum*—the presence of methylglyoxal synthase was proposed by Rodriguez et al. [10] as an important feature for bypassing the lower part of glycolysis, since it is an alternative catabolic pathway for triose phosphates. Glyoxalase family proteins have also been recognized as key components in *Planococcus halocryophilus* Or1, a non-spore forming *Firmicutes* isolated from subzero temperatures from high Arctic permafrost [12]. The breakdown of glyoxal and methylglyoxal, formed as by-products of several metabolic pathways under subzero conditions, allows the removal of reactive electrophilic species through the recycling of reactive carbonyls to lactate, which is further used for cellular metabolism [12]. These results were in line with previous work, showing that, in this bacterium, energy metabolism was repressed at −15 °C. However, the expression of succinic semialdehyde dehydrogenase, alcohol dehydrogenase and several oxidoreductases are increased, a fact that was attributed to the maintenance of energy metabolism and ATP levels [13]. A similar finding was reported in a transcriptomic analysis of *Pseudomonas extremaustralis*, a bacterium isolated from a temporary pond in Antarctica, in which the genes involved in primary metabolism were downregulated while those involved in ethanol oxidation—*exaA*, *exaB* and *exaC*—encoding a pyrroloquinoline quinone (PQQ)-dependent ethanol dehydrogenase, cytochrome c550 and an aldehyde dehydrogenase respectively, were upregulated [16]. Surprisingly, genetic and physiological approaches showed that this secondary pathway was essential for growth under low temperatures, as mutant strains in different ethanol oxidation related genes were unable to grow under cold conditions [16]. This result shows that, despite the lower energy yield in comparison with central metabolism, ethanol oxidation becomes essential under cold conditions where TCA and cytochrome coding genes are repressed [16].

In the marine bacterium *Sphingopyxis alaskensis*, which inhabits permanently cold waters (4–10 °C), the fatty acid metabolism influences energy generation since five enzymes of the fatty acid degradation pathway (β-oxidation) were more abundant at low temperatures, rendering acetyl-CoA, which can enter to the TCA [18]. These authors also observed that fatty acid enzymes have a role in the propionate pathway, where propionyl-CoA is the central metabolic intermediate. Propionyl-CoA can also be obtained from the catabolism of some aminoacids, and can be carboxylated to S-methylmalonyl-CoA and subsequently isomerized to succinyl-CoA, another intermediate of TCA [18]. Similarly, RNAseq studies of the well-characterized model strain *Pseudomonas putida* KT2440, growing at 10 °C, showed that the 2-methylcitrate pathway and branched aminoacid degradation were upregulated [21], allowing catabolization of propionate and propionyl-CoA, which are further transformed into 2-methylcitrate and later processed to succinate and pyruvate [22,23]. *P. putida* KT2440 is a derivative strain of the soil isolate *P. putida* mt-2 [24], which can grow at low temperatures [21,25,26]. Although this bacterium is not indigenous to cold environments, it shows similar adaptations, obtaining intermediate compounds for TCA replenishment by alternative pathways. This evidences the extent of this strategy for cold adaptation.

Overall, during cold growth, the glycolysis and TCA cycle seems to be repressed while other secondary pathways or the obtainment of intermediate compounds by alternative routes to bypass the complete pathway arise as important features for energy generation.

## 3. Compounds and Mechanisms Involved in Stress Resistance and Cold Adaptation

### 3.1. Compatible Solutes and Related Compounds

The importance of traits related to the metabolism of compatible solutes to cope with cold has been recognized in many cold-adapted bacteria. These compounds play an important role in osmoprotection and cryoprotection but can also function as carbon, nitrogen and energy sources [27]. The main studied compounds, glycine, betaine, glycerol, trehalose, sucrose, mannitol and sorbitol, can reduce the freezing point of the cytoplasm, prevent macromolecule aggregation, scavenge free radicals and stabilize cellular membranes under cold conditions [28].

Recently, genome comparison and global phenotypic characterization of two Antarctic strains of *Pseudoalteromonas*, *P. haloplanktis* TAC125 and *Pseudoalteromonas* sp. TB41, conducted at 4 °C and 15 °C revealed differences for cold adaptation. Remarkably, protein S-thiolation, regulated by glutathione and glutathionylspermidine, appeared to be a new possible mechanism for cold adaptation in TAC125 [29]. The study provided evidence of the relevance of glutathione metabolism in this bacterium that showed enhanced uptake (or catabolism) of compounds that may confer cryotolerance such as spermine, glutathione and ornithine [29]. These compounds probably link compatible solutes and oxidative stress resistance. In line with these findings, genome analysis of *Colwellia psychrerythraea* showed the presence of genes coding for proteins involved in the synthesis and the degradation of polyamides, nitrogen reserves polymers, considered as a probable unique adaptation to cold environments [27,30]. Moreover, metabolomic analysis of the Arctic isolate *Mesorhizobium* sp. Strain N33 revealed an increment in the accumulation of sarcosine, threonine and valine when this bacterium was grown at 4 °C, probably acting as cryoprotectants [31].

### 3.2. Polyhydroxyalkanoates Metabolism

Several bacteria indigenous to cold environments can synthesize polyhydroxyalkanoates (PHAs)—reserve polymers that have an important physiological role. These polymers are accumulated under unbalanced growth conditions, such as an excess of carbon source with respect to other nutrients such as nitrogen or phosphorus, acting as dynamic reservoirs of carbon and reducing equivalents [32]. PHAs endow bacteria with enhanced survival and resistance to a variety of environmental stress conditions while also having ecological relevance (e.g., [33,34,35,36]). Genome analysis of *C. psychrerythraea* revealed the capacity to produce PHA linked with its significant ability to produce and degrade fatty acids. Multiple gene duplications of acyl-CoA dehydrogenase and enoyl-CoA hydratase gene families were detected, probably indicating versatility in PHAs characteristics that it can synthesize [27]. Later proteomic analysis showed a significant increase in PHA depolymerase after 24 h at −10 °C, suggesting polymer utilization under freezing conditions [30].

In *S. alaskensis*, proteome analysis revealed an increased abundance of enzymes related to PHA synthesis at low temperatures, probably indicating that it may compensate for reduced rates of enzyme activity and nutrient transport, as an adaptive strategy to ensure that this pathway remains functional in the cold [18]. Among the proteins involved in PHA biosynthesis, phasins are the major PHA granule-associated protein that has multiple functions, and it has been proposed that it could play an active role in stress protection and fitness enhancement [37]. The PhaP phasin of *S. alaskensis*, along with a 3-hydroxybutyrate dehydrogenase that is presumably secreted, were found increased at low temperatures [18]. These findings suggested that the contribution of PHA metabolism to cold adaptation in *S. alaskensis* involves both the synthesis of PHA de novo and the secretion of enzymes suitable for scavenging extracellular PHA [18].

*P. extremaustralis* shows important traits related to the role of PHAs for low temperature adaptability. This bacterium produces a different kind of PHAs, short chain length PHAs (sclPHAs) as polyhydroxybutyrate (PHB) and medium chain length PHAs (mclPHAs) [38]. In particular, PHB production was found to be fundamental for cold growth and freezing survival [39]. PHB accumulation increased motility and the survival of planktonic cells in the biofilms developed by this bacterium under cold conditions, suggesting that the capability to accumulate PHB could constitute an adaptive advantage for the colonization of new ecological niches in those environments [40]. Interestingly, PHB production is not a common characteristic in *Pseudomonas* species, and genetic analysis of *P. extremaustralis* showed that PHB genes are located in a genomic island [38,41], highlighting the importance of lateral gene transfer events for adaptability. The relevance of the metabolism of this polymer for cold adaptation was found by analyzing a PHB synthase mutant strain of *P. extremaustralis* (*phbC*). The mutant was unable to grow at 10 °C and showed higher lipid peroxidation and a noticeable decrease in NADH/NAD ratio and NADPH content in comparison with the wild-type strain after a temperature downshift in which a rapid degradation of PHA was observed [39]. Protection conferred for PHA can lie in different non-excluding mechanisms. PHA metabolism is related to global stress responses, since an association between PHA degradation and the accumulation of the alarmone (p)ppGpp was found with a concomitant increase of the intracellular level of the master regulator RpoS [42,43]. In addition, a cryoprotective role was attributed to the PHB monomer, 3-hydroxybutyrate and it was observed that under freezing conditions PHB granules remain highly flexible, suggesting that they might protect bacterial cells against injury from intracellular and extracellular ice [44,45].

Regarding the PHA function in the cellular redox balance and the protection against oxidative stress derived from cold, some studies have reported that antioxidative enzymes (e.g., catalase, superoxide dismutase and glutathione peroxidase) are induced under cold conditions [46,47,48]. Some of these enzymes depend on nicotinamide dinucleotides as cofactors [49], giving an explanation regarding the PHA mechanism to cope with oxidative stress derived from cold exposure [32]. However, proteomic analysis performed in *P. haloplanktis* showed a repression of catalase, glutathione reductase and peroxiredoxin at low temperatures [15] and similar results were obtained in RNAseq experiments performed in *P. extremaustralis*, in which genes encoding the alkylhydroperoxidase, the glutathione peroxidase, the oxyR sensor and the superoxide dismutase were repressed at low temperatures in the early exponential growth phase [16]. Nevertheless, the adjustment of other pathways related to oxidative stress protection was found in these bacteria. The upregulation of *algZ* involved in alginate production and putrescine and spermidine accumulation genes was observed in concordance with the downregulation of iron acquisition, an important metabolic modification contributing to the alleviation of oxidative stress since iron is involved in the Fenton reaction [13,14,17]. In line with this, in *Psychrobacter* sp. PAMC 21119, putrescine synthesis was also found to be upregulated while heme protein synthesis was downregulated [20]. Putrescine and spermidine have been implicated in oxidative and cold stress responses [50,51].

In addition, screening for PHA producers was performed in different extreme cold environments. PHA producers belonging to *Pseudomonas* and *Janthinobacterium* genera, isolated from Antarctic soils, showed higher polymer accumulation between 5 °C and 20 °C in comparison with higher temperatures. The presence of highly unsaturated mclPHA in the isolate *Pseudomonas* sp. UMAB-40 was associated with its ability to survive in the cold [52]. PHA synthesis capability of bacterial isolates from Antarctic freshwater belonging to Alpha, Beta and Gammaproteobacteria was regarded as a common feature at pioneer sites [53]. Culture independent studies performed in extremely cold environments, such as the Baltic Sea and Greenland Sea ice, showed the presence of genes encoding PHA synthases in the bacterial community [54,55], suggesting the relevance of this metabolism for the whole community.

All these reports constitute evidence that points out the synthesis and degradation of PHA as an important trait for thriving under cold conditions.

## 4. Envelopes and Cold Adaptation

The composition and structure of envelopes are recognized as important for cold adaptation as they constitute a dynamic interface between cell and environment, enabling them to cope with environmental challenges. Among the strategies used by cold-adapted microorganisms, a general well-known mechanism consists in the modification of cell membrane lipid composition, favoring shorter chains and decreasing lipid saturation [4] in order to maintain membrane fluidity while avoiding stiffness at low temperatures. Although the modifications of the internal membrane are well studied (Table 1), little is known about the changes that occur in other envelope components.

In Gram negative bacteria, the envelope consists of an inner and an outer membrane separated by the periplasmic space containing a thin peptidoglycan layer. The outer membrane is formed by phospholipids, proteins and lipopolysaccharide (LPS). LPS contains the lipid A, anchored to the membrane, an intermediate oligosaccharide component (core) and an external O-polysaccharide. In *P. haloplanktis*, the presence of a higher percentage of cell envelope genes in the genome has been suggested as a specialized adaptation to cope with cold [56]. Similar observations were reported for *P. arcticus*, in which genes involved in membrane lipid and cell wall synthesis were identified as relevant for growth under subzero temperatures [57]. Proteomic analysis of *S. alaskensis* showed that several proteins related to cell wall, membrane, exopolysaccharide biosynthesis and envelope biogenesis had a higher abundance at 10 °C [18]. In the Antarctic bacterium *Pseudomonas syringae* Lz4w, it was observed that low temperatures caused changes in the composition and fluidity of LPS as higher polymyxin B sensitivity was detected along with an increased amount of hydroxy fatty acids [58]. In another Antarctic *Pseudomonas*—*P. extremaustralis*—the relevance of the core integrity of the LPS to cope with cold was evidenced by analyzing a *wapH* mutant strain, encoding a core LPS glycosyltransferase [59]. The *wapH* strain was impaired to grow in the cold. The *wapH* deficiency provoked a strong aggregative phenotype and modifications of envelope nanomechanical properties, such as lower flexibility and higher turgor pressure, cell permeability and surface area to volume ratio (S/V). Changes in these characteristics were also observed when *P. extremaustralis* was grown at different temperatures. These results indicate that LPS appears as a novel essential feature for active growth under cold conditions [59]. 

In Gram positive bacteria where the envelope is composed of the cell wall and the inner membrane, the importance of these components has also been recognized. The increase in cell wall biosynthesis transcripts was observed in *E. sibiricum* at −2.5 °C, suggesting that the thickening of the cell wall may protect the cell against disruption by ice formation and/or osmotic pressure, which can be generated at subzero temperatures [10]. Cell wall, membrane and envelope-component synthesis were also shown to be fundamental to supporting cold active growth in *Planococcus halocryophilus* Or1. This bacterium possesses an unusual cell envelope characterized by encrustations surrounding cells during subzero growth at −15 °C [13]. Transcriptomic analysis showed the relevance of genes encoding peptidoglycan synthetase, along with those involved in the synthesis of precursors [13]. Further microscopic characterization of the cell envelope showed increasing hydrophobicity and distinct extracellular encrustations of 20% calcium carbonate, 50% peptidoglycan and 29% choline closely associated with the cell wall. Along with this, genome analysis allowed the identification of several copies of genes encoding carbonic anhydrase, responsible for mineralization of calcium carbonate. The expression of a single copy of them, as well as genes related to peptidoglycan biosynthesis, was increased under low temperature conditions evidencing microbial-mediated calcium carbonate precipitation at subzero temperatures [60]. Another surprising feature of *P. halocryophilus* at sub-zero temperatures is that fatty-acid saturation increases with decreasing temperature [13]. It has been reported that *P. halocryophilus* employs an alternate mechanism for preserving membrane fluidity and that the fatty-acid desaturases are present but inactive at these temperatures [14].

Exopolysaccharides (EPS), which comprise a substantial component of the extracellular polymers surrounding bacterial cells, are also considered an important cellular function to cope with cold and ice [40,61]. *Pseudoalteromonas* sp., isolated from Arctic marine environments, produce a highly complex EPS with mannose as a main component, interestingly *P. haloplanktis* and other strains isolated from Antarctic environments also presented mannose as an important component of EPS [62,63,64]. These findings suggest that EPS production with specific characteristics is shared among cold-adapted bacteria. In addition, extracellular polymeric substances were related to the improvement of the freeze-thaw survival ratio in bacteria belonging to the *Winogradskyella*, *Colwellia* and *Shewanella* genera isolated from the Antarctic sponge [65], further indicating a cryoprotective role for these compounds.

Present evidence indicates the importance of the maintenance of membrane fluidity that appears as a common feature in cold-adapted bacteria but also highlights the relevance of several envelope components to cope with cold.

## 5. Functional Attributes of Cold-Adapted Bacterial Communities

In the next section, several cold bacterial community studies using global metagenome, metatranscriptome and metaproteome approaches are analyzed. The overall available information of these communities shows the constancy in general functional attributes involved in cold adaptation and the relevance of other metabolisms and features present in the complex microbial communities. Metagenome analysis of cyanobacterial mats from Arctic and Antarctic ice shelves showed that genes coding for functional responses to environmental stress such as exopolysaccharides, cold shock proteins and membrane modifications were found in all analyzed metagenomes [66]. The finding of a higher abundance of sequences (reads) matching the sigma B genes in the Antarctic mat was proposed as a feature reflecting more severe osmotic stress during freeze-up of the Antarctic ponds [66]. The results of six metagenomic datasets from different microbial mat communities of the perennially ice-covered Antarctic Lake Joyce were used to filter cold adaptation protein information, including antifreeze proteins, cold-shock DEAD-box protein A, cold-shock family of proteins (CSPs; including CspA, CspB, CspC, CspD, CspE and CspG), fatty acid desaturase, ice nucleation protein and trehalose synthase [67]. The analysis showed that while the cold-shock family of proteins, cold-shock DEAD-box protein A, antifreeze proteins, fatty acid desaturase and trehalose synthase were present in all mat samples at different levels, the ice nucleation protein was only found at very low levels in some of the samples [67], probably reflecting the presence of generalized and particular mechanisms for cold adaptation in the bacterial communities. Interestingly, a metagenomic study of the biofilm and planktonic microbial communities of an acid mine drainage stream located below ground at a low-temperature (6–10 °C) in Kristineberg mine (Sweden) also showed functional characteristics previously reported as related to cold growth. These functional characteristics include cold-shock and anti-freeze proteins and several compatible solutes production pathways, besides the typical functions related to pH homeostasis and metal resistance expected to thrive in this acidic metal-containing ecosystem [68]. A recent metagenomic work analyzed the bacterial communities of Alaskan Pleistocene permafrost over geologic time by exploring a chronosequence from 19,000 to 33,000 years before the present day [69]. The results showed shifts consistent with long-term survival strategies involving scavenging detrital biomass, horizontal gene transfer, chemotaxis, dormancy, environmental sensing and stress response, thus associating the increase of abundance of many genes with age with microbial community adaptation to extreme cryogenic environment [69]. A comparative metaproteomic study of winter and summer bacterioplankton from Antarctic Peninsula coastal surface waters showed differences in each season [70]. While in the summer, autotrophic carbon assimilation appears to be driven by oxygenic photoautotrophy, consistent with high light availability and intensity, during the dark winter chemolithoautotrophic metabolism involving the 3-hydroxypropionate/4-hydroxybutyrate cycle and the reverse tricarboxylic acid cycle represented the main processes [70]. Metatranscriptome analysis of microbial communities in Alaskan permafrost soils showed the dominance of stress responses, survival strategies and maintenance processes under frozen conditions, while upon thawing, a rapid enzymatic response to decomposing soil organic matter was observed as well as transcripts related to acetogenesis and heterotrophic methanogenic pathways utilizing acetate, methanol and methylamine [71].

The understanding of microbial communities is complex, because many metabolic groups coexist in the environment and much remains to be discovered. However, several studies have highlighted the presence of features recognized as relevant to coping with cold, such as cold shock proteins, compatible solutes, exopolysaccharides and membrane modifications. In addition, it was shown that permafrost communities may use a highly diverse and complex set of biochemical processes involved in carbon processing, organic matter decomposition, methane generation and oxidation and nitrogen cycling [69]. These findings provide new insights into microbial communities and their functions in Arctic environments impacted by climate change [69].

## 6. Conclusions

The available information regarding cold-adapted bacteria demonstrated that metabolic traits related to energy generation are adjusted during cold growth (Figure 1). These involve the use of secondary pathways or the generation of intermediate compounds to bypass the ROS-generating pathways (Table 1). In addition, a wide set of compounds and mechanisms involved in stress resistance are displayed, along with modifications in envelope components that arise as important novel features for cold adaptation (Figure 1). Reports analyzing the functioning of the whole community have also demonstrated the presence of the classical features recognized as relevant for coping with cold, such as cold shock proteins, compatible solutes, exopolysaccharides and membrane modifications (Figure 1) but also showed the presence of a highly diverse set of metabolic features.

## Figures and Tables

**Figure 1 life-08-00008-f001:**
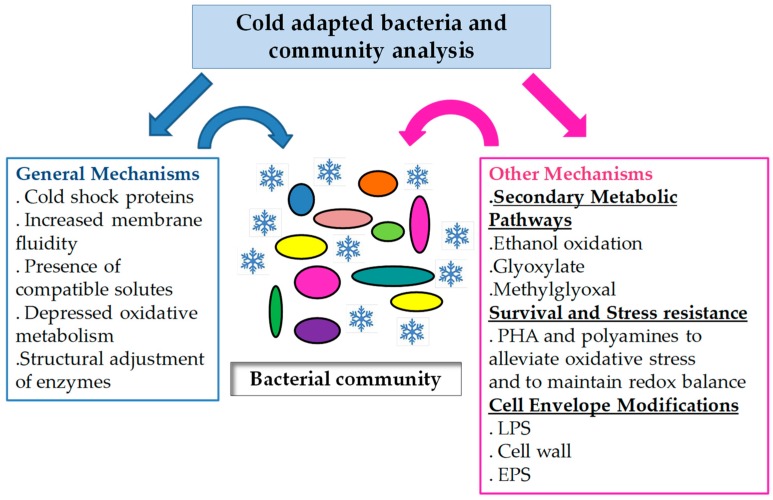
Mechanisms for cold adaptation. Available information regarding cold-adapted bacteria and microbial communities inhabiting cold environments shows that generalized and particular mechanisms could act in concert giving rise to cold adaptation profile of the entire community.

**Table 1 life-08-00008-t001:** Cold adaptive bacterial responses revealed by genomic, transcriptomic and proteomic studies.

Bacterial Species	Metabolic Features			Increase of Membrane Fluidity	References
	TCA Repression or Shortened	Cytochrome Repression	Presence of Alternative Pathways		
*Exiguobacterium sibiricum*	NI	X	X ^a^	X	[10]
*Nesterenkonia sp. AN1*	NI	NI	X ^a^	X	[11]
*Planococcus halocryophilus Or1*	X	X	X ^a^	Decrease	[12,13,14]
*Pseudoalteromonas haloplanktis*	X	X	NI	X	[15]
*Pseudomonas extremaustralis*	X	X	X ^c^	NI	[16]
*Psychrobacter arcticus 273-4*	X	NI	X ^a^	X	[17]
*Sphingopyxis alaskensis*	X	X	X ^b^	X	[18]

X: presence of the feature; NI: not informed; ^a^: Glyoxylate and methyglyoxal pathways; ^b^: Propionyl-CoA catabolism; ^c^: Ethanol oxidation; TCA: tricarboxylic acid cycle.

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
