# Peer review of "Reporting Key Features in Cold-Adapted Bacteria"

_life, 2018, doi:10.3390/life8010008_

Round 1
Reviewer 1 Report
The review paper by Tribelli and López deals with a very complex topic, which is really complicate to summarized. A great effort has been done by the Authors who have mainly based the review on recent data (more than an half of cited references have been published in the last 8 years, since 2010).
I find that all main arguments have been treated in the review, but some unbalances occur. For example, a lot of text is dedicated to PHA (the authors are experts in this) and energy generation, while other microbial features are often simply listed. I suggest to create sub-paragraphs to make the reading more “friendly”. In the case of EPS, two new papers have recently appeared (doi: 10.1128/AEM.01624-17 and doi: 10.1007/s11356-017-0851-z) on Antarctic bacterial isolates. Please, consider them (if you consider them useful) to enlarge the section of EPS which is currently scant.
I’d prefer the use of “cold-adapted” microorganisms as in my opinion it could be better to refer to both psychrophilic and psychrotrophic microbes.
Some typesetting errors occur, mainly in the use of comma (e.g. lines 80-83). The reference list need to be better formatted for the journal.
In my opinion, the manuscript deserves to be published in Life after minor revisions, mainly regarding the text formatting.
Author Response
Answers to Reviewer 1
R:I find that all main arguments have been treated in the review, but some unbalances occur. For example, a lot of text is dedicated to PHA (the authors are experts in this) and energy generation, while other microbial features are often simply listed. I suggest to create sub-paragraphs to make the reading more “friendly”. In the case of EPS, two new papers have recently appeared (doi: 10.1128/AEM.01624-17 and doi: 10.1007/s11356-017-0851-z) on Antarctic bacterial isolates. Please, consider them (if you consider them useful) to enlarge the section of EPS which is currently scant.
A: We have carefully revised the whole manuscript and we have also added subtitles in the section 3. We hope that this modification can help to improve reading. Energy generation and PHA metabolism are deeper analyzed because we have work on them but also because there is more information about these topics that we want to highlight as important features. In addition, we have included both papers concerning EPS in Antarctic bacteria as you suggested, that contribute to enrich EPS paragraph. In addition, we have deleted some paragraphs as the reviewer 2 suggested, so we think that the manuscript is now more balanced.
R: I’d prefer the use of “cold-adapted” microorganisms as in my opinion it could be better to refer to both psychrophilic and psychrotrophic microbes.
A: We have replaced the terms psychrophilic and psychrotrophic by cold adapted bacteria along the manuscript.
R: Some typesetting errors occur, mainly in the use of comma (e.g. lines 80-83). The reference list need to be better formatted for the journal.
A: We have carefully checked all the manuscript including the reference list and text formatting.
Reviewer 2 Report
General Comments
In general, I applaud this attempt by the authors to develop this theme. There is certainly new material in this review, and the approach does not duplicate recent reviews of a similar nature (references 2 – 6).
Sadly, the review is marred by being too long, and by the poor standard of English grammar throughout. Both can be solved without undue strain, firstly by increasing the focus rather (see below), and secondly by either using an editing service or bringing in an English-speaking advisor/co-author.
Specific Comments
1. The title is cumbersome: one of the two words ‘novel’ and ‘key’ can be removed, and the term ‘indigenous’ is redundant.
2. There is a possible distinction in physiology and adaptation between true psychrophiles and so-called psychrotolerant organisms. The former, most of which are marine and aquatic (or permafrost), tolerate consistently low temperatures with little temperature variation. The latter (most Antarctic soil organisms, for example) experience wildly fluctuating temperatures, where the both the upper and lower extreme temperatures are quite extreme, and where high stress events , such as freeze-thaw, are common. The responses and adaptations in these two groups of organisms are likely to be very different, although I am not sure that there are many comprehensive studies addressing this issue. The authors should at least demonstrate an awareness of the issue: at present, the ‘adaptations’ of both groups are completely intermingled.
3. There is a tendency, which is very strongly repeated throughout this review, to make the untested assumption that any change (be it genomics, transcriptomics or proteomics data) which is observed in a psychrophilic organism is automatically a ‘psychrophilic’ characteristics. While I acknowledge that the comparative data may not exist in many instances, the authors should be very cogniscent of this issue and highlight it where appropriate. Certainly, some of the ‘psychrophilic’ adaptations noted are general stress-response effects, and some of the responses are also seen in other extreme (but not low temperature) systems.
4. In general the text is too long. The authors should attempt to shorten and focus the text, by removing much of the older literature and discussions around such (which has been reviewed many times), by removing redundant paragraphs (I specifically refer to lines 339 – 352) but other sections might be considered, and by removing section 6 (which is interesting, but a diversion from the core theme).
Author Response
General Comments
R: Sadly, the review is marred by being too long, and by the poor standard of English grammar throughout. Both can be solved without undue strain, firstly by increasing the focus rather (see below), and secondly by either using an editing service or bringing in an English-speaking advisor/co-author.A: We have asked for the help of a researcher proficient in English and we have tried to emphasize the focus of the work. The performed changes are described in detail below.
Specific Comments
R1. The title is cumbersome: one of the two words ‘novel’ and ‘key’ can be removed, and the term ‘indigenous’ is redundant.
A: We have modified the title by removing the words “novel” and “indigenous”.
R2. There is a possible distinction in physiology and adaptation between true psychrophiles and so-called psychrotolerant organisms. The former, most of which are marine and aquatic (or permafrost), tolerate consistently low temperatures with little temperature variation. The latter (most Antarctic soil organisms, for example) experience wildly fluctuating temperatures, where the both the upper and lower extreme temperatures are quite extreme, and where high stress events, such as freeze-thaw, are common. The responses and adaptations in these two groups of organisms are likely to be very different, although I am not sure that there are many comprehensive studies addressing this issue. The authors should at least demonstrate an awareness of the issue: at present, the ‘adaptations’ of both groups are completely intermingled.
A: We agree with the reviewer that this is a very interesting point, but it is difficult to clarify. In fact, the separation of permafrost bacteria as “subzero inhabitants” from those found in other “perhaps” more variable environments regarding temperature, seems logical. However, it is very difficult to perform an accurate separation because several permafrost bacteria can also grow above freezing temperatures as P. halocryophilus Or1, several Psychrobacter spp. and Exiguobacterium and several marine bacteria thrive in permanent cold environments despite they can grow at higher temperatures. In our opinion it is a topic that still deserves more investigation. Then, bacteria were arranged by metabolic similarity and we try to maintain a brief reference to the isolation place and temperature, as a register for future studies. For this reason, we agreed with the concepts of Cavicchioli (2016) regarding this complex subject and used the term psychrophile to bacteria native from cold environments. However, we have now changed the term psychrophile by "cold adapted" in the new version of the manuscript, as the reviewer 1 suggested.
R3. There is a tendency, which is very strongly repeated throughout this review, to make the untested assumption that any change (be it genomics, transcriptomics or proteomics data) which is observed in a psychrophilic organism is automatically a ‘psychrophilic’ characteristics. While I acknowledge that the comparative data may not exist in many instances, the authors should be very cogniscent of this issue and highlight it where appropriate. Certainly, some of the ‘psychrophilic’ adaptations noted are general stress-response effects, and some of the responses are also seen in other extreme (but not low temperature) systems.
A: This is another important point in which much knowledge is still lacking. We try to focus on new features that authors find associated with organisms adapted to cold environments. However, it is true that several characteristics involve general stress responses and although there is not information about all cold adapted bacteria that were analyzed, some of them are recognized as polyextremophiles as is the case of the Antarctic Nesterenkonia. Several stress resistance mechanisms against different stress agents have been also analyzed in P. extremaustralis. Considering this observation, we have checked all the manuscript trying not to make generalization or incorrect assumptions.
R4. In general the text is too long. The authors should attempt to shorten and focus the text, by removing much of the older literature and discussions around such (which has been reviewed many times), by removing redundant paragraphs (I specifically refer to lines 339 – 352) but other sections might be considered, and by removing section 6 (which is interesting, but a diversion from the core theme).
A; We have revised all the text trying to shorten it. We have focused in the older literature as the reviewer suggested. Several sentences were deleted including most of lines 339-352 of the previous version of the manuscript. However, older references and associate discussion were not too much, because we conceived the manuscript taking in mind do not repeat information that had been extensively revised in previous works, thus concentrating our efforts in recent works. Even so, we have carefully checked all the manuscript and have avoided include some general information. Regarding section 6 we have strongly reduced the content but we rather to maintain it because the topic of the special issue of Life is “Extremophiles and the origin of life”, we think that could be valuable to link cold extremophile features and origin of life.
Round 2
Reviewer 2 Report
The authors have responded reasonably comprehensively to the comments and suggestions in my first review. However, in general, their response has been fairly muted and fairly shallow. The product is a review which is better than the first version, but which still falls somewhat short of the ideal.
The core of the issue lies in the title - 'Unravelling'....but in reality, the review does not really unravel anything - the term 'reporting' would be more appropriate. In essence, the review does not really deliver any substantively novel insights into the mechanisms of cold adaptation, but it does provide a useful and comprehensive assembly of a reasonably large volume of relevant literature.
The standard of English is better, but still a long way from perfect - so this is an issue for the journal's policy - whether the text in this manuscripts passes the journal's mimimum standards for grammatical quality.
I reiterate one of my earlier points - the final section (6) could be deleted completely without loss to the review. It is actually not relevant to the core theme of the review.
Author Response
Answers to reviewer 2
R: reviewer comment, A: answer
R: The core of the issue lies in the title - 'Unravelling'....but in reality, the review does not really unravel anything - the term 'reporting' would be more appropriate. In essence, the review does not really deliver any substantively novel insights into the mechanisms of cold adaptation, but it does provide a useful and comprehensive assembly of a reasonably large volume of relevant literature.
A: We have changed the word “Unraveling” by “Reporting” in the title as the reviewer suggested.
R: The standard of English is better, but still a long way from perfect - so this is an issue for the journal's policy - whether the text in this manuscripts passes the journal's mimimum standards for grammatical quality.
A: We have carefully revised the English of the manuscript with help of a person proficient in English. We hope that it can pass the journal´s standards quality.
R: I reiterate one of my earlier points - the final section (6) could be deleted completely without loss to the review. It is actually not relevant to the core theme of the review.
A: We have deleted the section 6 in the new version of the manuscript.
